# The Effects of Repeated Kurome Treatment on Chinese Lacquer and Its Film Properties

**DOI:** 10.3390/polym17111481

**Published:** 2025-05-27

**Authors:** Jiangyan Hou, Yao Wang, Tianyi Wang, Guanglin Xu, Xinhao Feng, Xinyou Liu

**Affiliations:** 1College of Furnishing and Industrial Design, Nanjing Forestry University, Str. Longpan No. 159, Nanjing 210037, China; houjiang@njfu.edu.cn (J.H.); 2381132433@njfu.edu.cn (Y.W.); 13861210436@njfu.edu.cn (T.W.); xuguanglinwfxy@163.com (G.X.); fengxinhao@hotmail.com (X.F.); 2Co-Innovation Center of Efficient Processing and Utilization of Forest Resources, Nanjing Forestry University, Nanjing 210037, China

**Keywords:** chinese lacquer (urushi), kurome treatment, oxidative polymerization, viscosity modulation, curing time optimization, lacquer film performance

## Abstract

This study systematically investigates the effects of repeated Kurome treatment—a physical modification method combining mechanical stirring and oxidative regulation—on the processing characteristics and film properties of Chinese lacquer (urushi). By subjecting raw lacquer to 1–4 cycles of hydration–dehydration (KL1–KL4), the researchers observed a significant increase in viscosity (from 12,688 to 16,468 mPa·s) and a dramatic reduction in curing time (from 74 h to just 3.6 h), driven by deep oxidation of urushiol and quinone-mediated crosslinking, as confirmed by FTIR spectroscopy. The Kurome treatment enabled controlled darkening (L* value decreased from 29.31 to 26.89) while maintaining stable hue and gloss (88.96–90.96 GU), with no adverse effects on abrasion resistance (mass loss of 0.126–0.150 g/100 r) or adhesion (9.58–9.75 MPa). The reduced transparency of the KL3/KL4 films is associated with a densified polymer network, a feature that may benefit protective coatings. Scanning electron microscopy (SEM) analysis confirmed the formation of uniform, defect-free surfaces across all treatment groups. Among them, the KL2 group (viscosity of 14,630 mPa·s, curing time of 9.2 h) exhibited the most favorable balance for industrial applications. This study establishes Kurome technology as a low-carbon, additive-free strategy that enhances the processability of Chinese lacquer while preserving its traditional craftsmanship standards, offering scientific support for its sustainable use in modern coatings and cultural heritage conservation.

## 1. Introduction

Chinese lacquer (urushi), derived from the sap of *Toxicodendron vernicifluum*, is one of the oldest known sustainable coatings in human history. Its unique composition—primarily urushiol, laccase, polysaccharides, and water—endows it with exceptional corrosion resistance, abrasion durability, and antioxidative properties [1,2], earning it the reputation of the “king of coatings”. Since the Neolithic era, Chinese lacquer has served as a key material medium in East Asian civilizations [3]. Today, millions of lacquerware artifacts preserved in museums around the world testify to its central role in the global cultural heritage.

However, the film-forming process of Chinese lacquer relies on a laccase-catalyzed oxidative polymerization, which requires specific environmental conditions: a temperature range of 20–30 °C and relative humidity between 70–85% [4,5,6]. These stringent curing conditions result in an extended drying period, often lasting weeks or even months, thereby significantly limiting its viability for modern industrial applications [7].

To address this challenge, prior research has explored various modification strategies such as chemical driers and ultraviolet curing [8,9,10]. However, these approaches often pose environmental hazards or compromise the mechanical properties of the cured films [11,12]. Kurome treatment, a physical modification method that involves mechanical stirring combined with oxidative regulation, offers a promising alternative [13]. It accelerates the polymerization of urushiol without the need for additives, making it both eco-friendly and cost-effective. Although several studies have investigated Kurome treatment, they have predominantly examined single-cycle applications [14,15,16], providing limited insight into how repeated treatment cycles may influence the lacquer’s physicochemical properties and film performance. In contrast, the present study systematically evaluates the impact of multiple Kurome cycles and identifies two treatment rounds as optimal. This novel approach expands the understanding of Kurome processing and offers a new pathway for enhancing the performance of natural lacquer coatings.

This study presents the first systematic investigation into the effects of repeated Kurome treatment on the processing characteristics of Chinese lacquer (e.g., viscosity and FTIR spectral evolution) as well as the comprehensive performance of the resulting films (e.g., color, transparency, surface roughness, gloss and adhesion strength). The goal is to establish a theoretical foundation for the low-carbon and efficient modification of urushi, thereby promoting its innovative applications in modern coating technologies and heritage conservation.

## 2. Materials and Methods

### 2.1. Materials

The raw Chinese lacquer used in this study was harvested from *Toxicodendron vernicifluum* trees in Maoba, Enshi, Hubei Province, China, in May 2023. After collection, the raw lacquer was filtered through a 300-mesh nylon sieve to remove impurities and stored in light-proof containers at 4 °C. The solid content was measured at 63.27 ± 0.84%.

Standard glass slides (76.2 mm × 25.4 mm × 1.2 mm) were ultrasonically cleaned sequentially with acetone and deionized water for 10 min, then dried and stored in a desiccator for later use [17].

Radial-Cut Wood Boards: Knot-free, evenly grained Chinese fir (*Cunninghamia lanceolata*) boards (moisture content 10 ± 2%) were cut into 100 mm × 100 mm × 10 mm specimens. Surfaces were sanded with 400- and 800-grit sandpapers to achieve a surface roughness (Ra) ≤ 1.2 μm, wiped clean with anhydrous ethanol, and conditioned for 14 days at 20 °C and 55% relative humidity to reach equilibrium [18].

### 2.2. Kurome Treatment Process

Initial dehydration was conducted by placing 100 g of raw lacquer into a 500 mL glass beaker, followed by stirring at 300 rpm and 40 °C using a thermostatic magnetic stirrer (IKA RCT Basic, ±0.5 °C accuracy). Water content was measured every 30 min until three consecutive measurements stabilized at 5%. The dehydration was followed by cyclic Kurome treatments: distilled water was added to the dehydrated lacquer at a mass ratio of 1:0.3 (e.g., KL1 treatment: 100 g lacquer + 30 g water) [19]. The mixture was stirred under the same conditions until the water content again decreased to 5%. After each cycle, the lacquer was degassed in a vacuum defoaming machine (0.08 MPa, 25 °C) for 5 min. This hydration–dehydration cycle was repeated 1–4 times to obtain samples labeled KL1 through KL4. All treated samples were sealed and stored in a 25 °C incubator under light-proof and constant humidity conditions [20,21,22].

### 2.3. Characterization of Chinese Lacquer Properties

#### 2.3.1. Viscosity Measurement

A rotational viscometer (NDJ-8S, Shanghai Fangrui Instruments, Shanghai, China) was used after equilibrating samples in a 25 ± 0.1 °C water bath for 30 min. Using rotor No. 4 (diameter: 18.72 mm; height: 65 mm, TechSpin Inc., Austin, TX, USA) at 6 rpm, the torque value was recorded after stabilization, and dynamic viscosity (mPa·s) was calculated. Each sample was tested 5 times, and the average was taken after removing outliers.

#### 2.3.2. Curing Time Determination

According to GB/T 1728-2020 [23], lacquer films (wet thickness: 150 ± 5 μm) were prepared using a wire-wound applicator to ensure uniform thickness, and then cured in a constant temperature–humidity chamber (25 ± 1 °C, 80 ± 2% RH). Surface drying time was assessed using the finger-touch method: clean filter paper was used to lightly touch the film every 30 min. The point at which no adhesion or fingerprint remained was recorded. Full curing time was defined as the time required for the film to reach a pencil hardness of 2H (measured according to GB/T 6739-2006) [24].

#### 2.3.3. Fourier Transform Infrared Spectroscopy (FTIR)

Uncured lacquer (0.5 mg) was mixed with 200 mg of KBr powder (Merck KGaA, Darmstadt, Germany), ground evenly in an agate mortar, and pressed into transparent pellets (15 MPa, held for 2 min). FTIR spectra were obtained using a Nicolet iS50 spectrometer (Thermo Fisher Scientific, Waltham, MA, USA) over 400–4000 cm^−1^ with a 4 cm^−1^ resolution and 32 scans.

### 2.4. Film Performance Tests

Processed lacquer was applied to glass slides (for transparency and SEM tests) and wood panels (for other performance tests) using a wire-wound film applicator (BYK 5350, 50 ± 1 μm wet thickness, BYK Additives & Instruments (part of ALTANA Group, Wesel, Germany). All films were cured in a 25 ± 1 °C, 80 ± 2% RH environment for 21 days. Temperature and humidity were monitored every 24 h to ensure stability [25].

#### 2.4.1. Color and Gloss

Color was measured using a CM-2600d spectrophotometer (Konica Minolta, Inc., Tokyo, Japan) under a D65 light source and a 10° observer angle. L* (lightness), a* (red-green), and b* (yellow-blue) values were recorded, and color difference ΔE was calculated relative to untreated lacquer (KL0): $\Delta E=\sqrt{\Delta L^{2}+\Delta a^{2}+\Delta b^{2}}$.

Gloss was measured with a BYK-micro-TRI-gloss meter (BYK-Gardner, Wesel, Germany) at a 60° incident angle across five different points; the average was reported in gloss units (GU) [26].

#### 2.4.2. Transparency

Transparency was assessed at 550 nm using a UV-2600 UV-Vis spectrophotometer (Shimadzu, Kyoto, Japan), with clean glass slides as reference. Lacquer films with a uniform wet thickness of 50 ± 1 μm were prepared using a wire-wound film applicator to ensure consistency. Scan range: 300–800 nm; optical path difference corrected to ±0.2%.

#### 2.4.3. Surface Roughness

Surface roughness was evaluated with a MarSurf M300 profilometer (Mahr, Göttingen, Germany), using a 2 μm radius stylus over a 4.8 mm scan length. Parameters R_a_ (arithmetical mean deviation) and R_z_ (maximum height of profile) were recorded. Ten parallel lines were measured per sample, spaced 2 mm apart, avoiding edge effects [27].

#### 2.4.4. Abrasion Resistance

In accordance with GB/T 1768-2006 [28], a Taber 5135 abrasion tester (Taber, North Tonawanda, NY, USA) with CS-10 wheels (50 mm diameter, medium hardness, Taber, North Tonawanda, NY, USA) was used. Load: 500 g; speed: 60 rpm; 100 cycles. Film weight was measured before and after testing using an electronic balance (accuracy: 0.1 mg) (Beiheng, Beijing, China )to calculate mass loss.

#### 2.4.5. Adhesion Strength

Adhesion strength was evaluated in accordance with ASTM D4541 [29] using a PosiTest AT-M pull-off adhesion tester (DeFelsko, Ogdensburg, NY, USA). Defect-free areas on the lacquer film were selected and lightly abraded with 400-grit sandpaper, then cleaned with isopropanol to prepare the surface. Aluminum dollies (20 mm in diameter, Wotefeier, Shenzhen, China) were bonded to the film using a two-part epoxy adhesive (Araldite 2014, Huntsman, Basel, Switzerland), ensuring an adhesive layer thickness of 50 ± 10 μm. The assemblies were cured at 25 °C and 50 ± 5% relative humidity for 24 h. The pull-off test was conducted by applying a tensile load at a rate of 0.5 MPa/s until the film detached, and the maximum tensile strength (MPa) was recorded. Fracture surfaces were examined both visually and under a microscope to classify failure modes as cohesive (within the film), adhesive (at the film–substrate interface), or mixed (≥50% interfacial failure). Five replicate tests were performed for each sample, and results with deviations exceeding 15% were excluded. The final adhesion strength values were averaged and reported with an accuracy of 0.1 MPa.

#### 2.4.6. Surface Morphology (SEM)

Lacquer films were fractured in liquid nitrogen, sputter-coated with gold (Hitachi E-1045, 10 nm thickness, Hitachi, Tokyo, Japan), and examined with a SU8010 field-emission scanning electron microscope (Hitachi, Tokyo, Japan) at 5 kV accelerating voltage and 8 mm working distance. The surface morphologies were observed at magnifications of 500× to 5000× [30,31].

### 2.5. Statistical Analysis

All measurements, including the viscosity and curing time of raw lacquer as well as the color, gloss, surface roughness, abrasion resistance, and adhesion strength of the lacquer films, were conducted using five parallel samples per group to calculate the mean values. One-way analysis of variance (ANOVA) was performed using SPSS 26.0 to assess the significance of differences among groups. Homogeneity of variance was tested prior to ANOVA, with a significance level set at *p* < 0.05 [32].

## 3. Results and Discussions

### 3.1. Changes in Chinese Lacquer Properties

Experimental data in Table 1. revealed that with increasing cycles of Kurome treatment (from KL0 to KL4), the physicochemical properties of Chinese raw lacquer changed markedly, particularly in terms of viscosity and curing behavior. Specifically, the viscosity increased significantly from 12,688 mPa·s in the untreated sample (KL0) to 16,468 mPa·s in KL4 (*p* < 0.001), which can be directly attributed to enhanced oxidative polymerization of urushiol [33]. However, the viscosity increase plateaued between KL3 and KL4 (Δ = 1444 mPa·s), suggesting that the system may have approached a state of dynamic equilibrium in polymerization [34,35].

Meanwhile, Kurome treatment also greatly accelerated lacquer drying. Surface drying time dropped from 234 min (KL0) to 21 min (KL3), and full curing time from 74 h to just 3.0 h. This dramatic enhancement can be attributed to reduced moisture content improving laccase activity [36], partial prepolymer formation decreasing oxidation steps [37], and more uniform films facilitating oxygen diffusion [38]. However, an unexpected increase in both surface drying time (from 21 min to 24 min) and full curing time (from 3.0 h to 3.6 h) was observed in KL4. This deviation is not simply a plateau effect, as the viscosity continued to rise. Rather, it may result from excessive polymerization leading to internal stress accumulation within the film. These stresses can densify the microstructure, reducing oxygen permeability and impairing enzymatic crosslinking in the final curing stage [39]. Since curing is an oxygen-dependent process, hindered diffusion could partially counteract the benefits of increased prepolymer content.

From a practical standpoint, KL2 offered the best balance between moderate viscosity (14,630 mPa·s) and efficient curing (9.2 h), showing performance comparable to certain modern industrial coatings [40]. In contrast, although KL3 and KL4 exhibited faster initial curing, their higher viscosities and potential curing instability suggest diminishing returns, indicating the need for careful optimization—possibly through gradient treatment strategies [41]. It is worth noting that our study did not directly measure laccase activity or polymer molecular weight distribution. These aspects, which are critical for confirming the proposed mechanisms, should be addressed in future work using enzyme inhibition assays and gel permeation chromatography (GPC) [42,43].

Kurome significantly reshaped the chemical structure of Chinese lacquer by dynamically regulating the oxidative cross-linking pathway of urushiol (Figure 1). Infrared spectroscopy analysis revealed a continuous enhancement of the carbonyl (C=O) absorption peak at 1730 cm^−1^, directly confirming the laccase-catalyzed transformation of catechol into quinone structures [44], which directly confirms the laccase-catalyzed transformation of catechol into quinone. The peaks at 2925 and 2855 cm^−1^, attributed to aliphatic C–H stretching vibrations (e.g., –CH_2_– and –CH_3_ groups in urushiol’s side chains), exhibited gradual attenuation, suggesting oxidative modification of the alkyl chains. The simultaneous weakening of absorption peaks in the 1595–1646 cm^−1^ range (C=C skeletal vibrations) and at 1475 cm^−1^ (C–H bending) indicates the disruption of the aromatic conjugation system, while the diminished peak at 3010 cm^−1^ (aromatic C–H stretching and/or residual O–H vibrations from catechol) reflects reduced electron density of the benzene ring due to oxidation [45,46,47].

Notably, the broad peak at 3360 cm^−1^, primarily assigned to moisture in untreated samples, significantly decreased after Kurome treatment, consistent with partial drying during processing. To further isolate the intrinsic O–H contributions from catechol, control experiments with desiccator-dried samples were performed, confirming that residual O–H vibrations near 3350 cm^−1^ persisted, corresponding to unoxidized catechol moieties.

In terms of substitution patterns and bond type reconfiguration, the continuous decline of peaks at 982 cm^−1^ (ortho-substituted C–H), 1280 cm^−1^ (symmetric C–O–C stretching), and 1186 cm^−1^ (asymmetric C–O–C stretching) suggests the transformation of catechol hydroxyl groups into carbonyls and the transition from ether bonds (C–O–C) to more stable C–C bonds [48]. The reduction at 1016 cm^−1^ (C–O stretching) further confirms hydroxyl group consumption in condensation reactions. These chemical evolutions collectively facilitated the formation of urushiol prepolymers and the densification of the cross-linked network, which macroscopically manifested as increased viscosity (KL0 → KL4: 12,688 → 16,468 mPa·s) and shortened drying time (full dry time from 74 → 3.6 h) [49]. However, the further decline of the 982 cm^−1^ peak at the KL4 stage (with viscosity still increasing but full dry time rebounding to 3.6 h) suggests that excessive cross-linking leads to increased molecular chain rigidity and internal stress accumulation, thereby limiting deep curing kinetics [50], which aligns with the “brittleness-toughness balance” effect observed in highly cross-linked systems.

### 3.2. Variations in Chinese Lacquer Film Properties

Table 2 summarizes the performance parameters of Chinese lacquer films under different Kurome treatment processes, including color, gloss, surface roughness, abrasion resistance, and adhesion. Overall, Kurome treatment effectively adjusts tonal depth while maintaining high structural integrity without significantly compromising the film’s visual consistency or mechanical stability.

Color measurements revealed a progressive decline in film lightness (L* value) with increasing Kurome treatment intensity. The KL3 (26.93) and KL4 (26.89) groups exhibited markedly lower lightness compared to untreated KL0 (29.31), with statistically significant differences (*p* < 0.001). This indicates that Kurome treatment induces darkening of the film color, attributable to intensified urushiol oxidation and increased quinone structure formation during processing, which enhances visible light absorption [51,52]. Despite pronounced changes in lightness, other color parameters (a, b, C, ΔE) showed no significant variations (*p* > 0.05), confirming stable hue and saturation. This finding holds positive implications for visual restoration and aesthetic preservation of traditional lacquerware, demonstrating Kurome’s applicability in scenarios requiring strict color uniformity.

Kurome treatment did not significantly alter surface gloss (*p* = 0.157), with gloss values consistently ranging between 88.96 GU and 90.96 GU, indicating high gloss retention. This suggests that the formation and cross-linking of urushiol prepolymers preserved surface homogeneity and minimized microscale roughness-induced light scattering [53]. High gloss remains a critical quality indicator for premium lacquer films, particularly in decorative crafts and functional artworks requiring both aesthetic appeal and contamination resistance [54].

Average roughness (R_a_) ranged narrowly from 0.122 to 0.138 μm, while maximum peak-to-valley height (R_z_) varied between 1.487 and 1.564 μm. Statistical analysis showed no significant intergroup differences (R_a_: *p* = 0.581; R_z_: *p* = 0.088), though the borderline significance of Rz (*p* = 0.088) suggests potential Kurome-induced microstructural modifications, particularly in high-intensity treatment groups (KL3/KL4). This trend may arise from increased prepolymer particle size and altered flow dynamics, leading to localized thickness variations [55].

Mass loss across groups (0.126–0.150 g per 100 r) showed no statistical significance (*p* = 0.262), confirming that Kurome treatment preserves the inherent abrasion resistance of Chinese lacquer. This durability stems from its densely crosslinked natural polymer network [56], demonstrating that accelerated oxidation and curing processes can achieve process optimization without sacrificing wear resistance.

Adhesion values remained stable at 9.58–9.75 MPa with no significant differences (*p* = 0.551), proving Kurome treatment does not compromise interfacial bonding. Strong adhesion arises from urushiol’s effective wetting and penetration into wood substrates during early oxidation stages, forming robust chemical or physical interlocking [57]. This is critical for enhancing the long-term durability of lacquerware in high-usage applications such as furniture or ceremonial artifacts.

Kurome treatment also influenced optical transparency (Figure 2). Untreated KL0 displayed strong translucency, whereas KL3 and KL4 films transitioned toward semi-opacity or near-opacity. This phenomenon correlates with increased scattering from densified polymeric networks formed under high oxidation states [58]. While reduced transparency may affect decorative applications requiring visual depth, it benefits protective coatings or opaque layers in heritage conservation.

SEM observations (Figure 3) revealed uniformly dense film surfaces across treatment groups, devoid of particles, cracks, or phase separation. This confirms that Kurome-modified lacquer retains excellent flowability and spreading capacity during curing, even at elevated polymerization levels and viscosities, ensuring structural integrity.

## 4. Conclusions

This study demonstrates that repeated Kurome treatment effectively enhances the polymerization of Chinese lacquer (urushi) by promoting urushiol oxidation and prepolymer formation, as evidenced by FTIR spectral evolution (e.g., intensified C=O stretching at 1730 cm^−1^) and viscosity increases from 12,688 to 16,468 mPa·s (*p* < 0.001). While KL2 achieved an optimal balance between viscosity (14,630 mPa·s) and curing efficiency (9.2 h full curing), rivaling modern coatings, excessive treatment (KL3/KL4) induced over-polymerization, slightly prolonging curing times due to internal stress accumulation. The process significantly darkened film color (ΔL = 2.42–2.43, *p* < 0.001) through quinone formation, yet preserved chromatic stability (a, b, ΔE: *p* > 0.05), aligning with heritage conservation needs. Critically, Kurome treatment maintained high gloss (88.96–90.96 GU), low roughness (Ra: 0.122–0.138 μm), abrasion resistance (0.126–0.150 g/100 r loss), and adhesion strength (9.58–9.75 MPa), with SEM confirming uniformly dense, defect-free microstructures. Reduced transparency in KL3/KL4 films, attributed to light scattering from densified networks, offers advantages for protective coatings despite limiting decorative translucency. Mechanistically, cyclic hydration–dehydration enhances laccase activity and prepolymer formation, accelerating curing without additives—an eco-friendly strategy bridging traditional craftsmanship and industrial demands. Future work should explore molecular weight dynamics (via GPC), hybrid additive-Kurome treatments, and environmental durability testing to further optimize urushi’s role in sustainable manufacturing and cultural preservation, reaffirming its status as the timeless “king of coatings”.

## Figures and Tables

**Figure 1 polymers-17-01481-f001:**
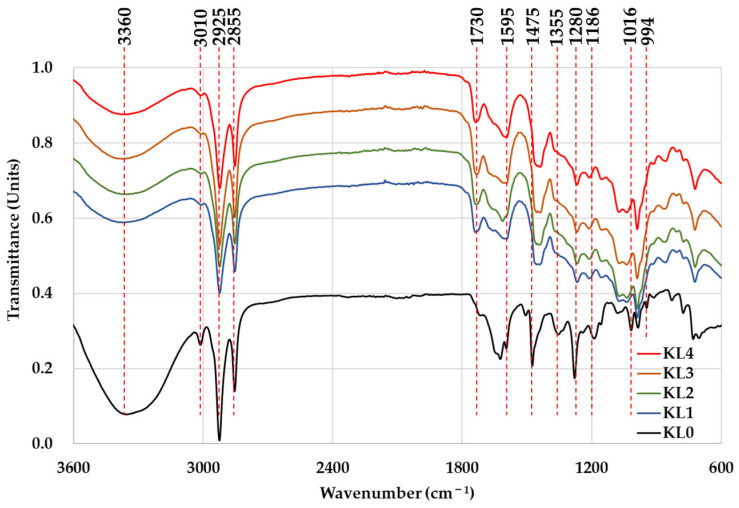
Comparative FTIR spectra of Chinese lacquer under different Kurome treatment conditions.

**Figure 2 polymers-17-01481-f002:**
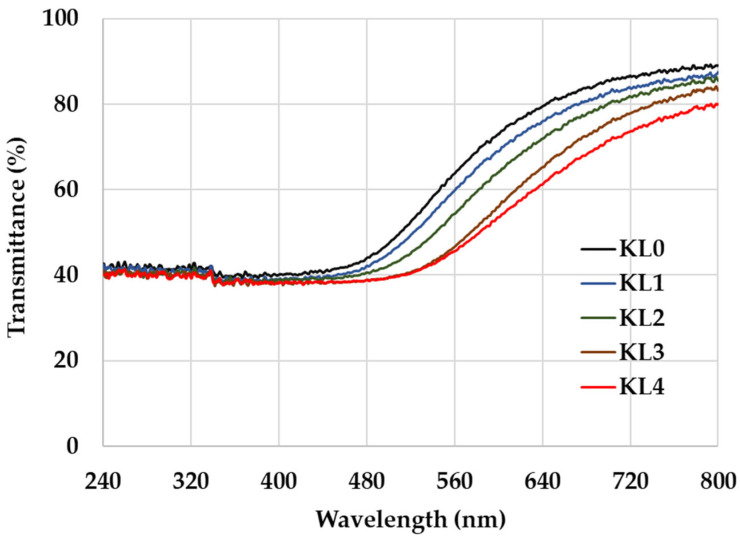
Light transmittance of Chinese lacquer films under different Kurome treatment processes.

**Figure 3 polymers-17-01481-f003:**
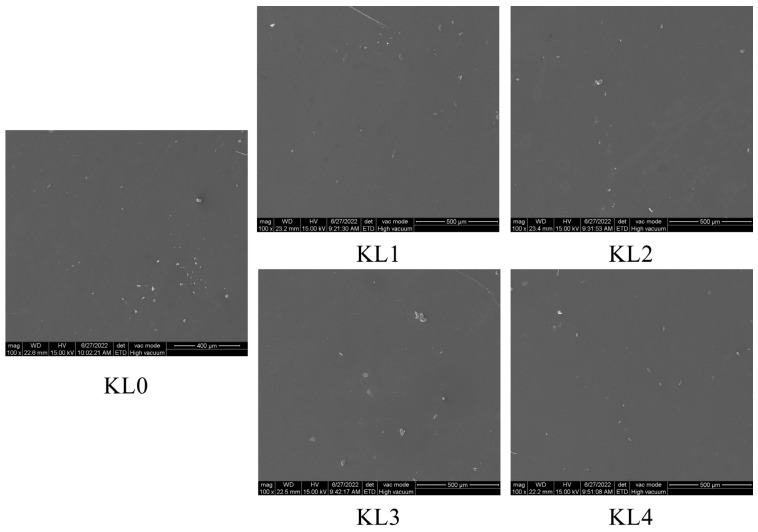
SEM micrographs of Chinese lacquer films under different Kurome treatment processes.

**Table 1 polymers-17-01481-t001:** Viscosity and curing time of Chinese lacquer under different Kurome treatments.

Chinese Lacquer	Viscosity (mPa·s)	Surface Dry Time (min)	Full Curing Time (h)
KL0	12,688 ± 666	234 ± 44	74 ± 13
KL1	13,520 ± 532	168 ± 34	48 ± 14
KL2	14,630 ± 583	41 ± 14	9.2 ± 2.7
KL3	15,024 ± 963	21 ± 6	3.0 ± 0.9
KL4	16,468 ± 475	24 ± 8	3.6 ± 0.6
*p* values	<0.001	<0.001	<0.001

**Table 2 polymers-17-01481-t002:** The performance parameters of Chinese lacquer films under different Kurome treatment processes.

Chinese Lacquer	Color Measurement					Gloss (GU)	Roughness		Mass Loss (g/100 r)	Adhesion (MPa)
	L	a	b	C	ΔE		Ra (μm)	Rz (μm)		
KL0	29.31 ^a^	2.98 ^a^	0.53 ^a^	3.03 ^a^	0	88.96 ^a^	0.132 ^a^	1.487 ^a^	0.134 ^a^	9.58 ^a^
KL1	29.12 ^a^	2.97 ^a^	0.64 ^a^	3.04 ^a^	0.22	88.98 ^a^	0.138 ^a^	1.549 ^a^	0.136 ^a^	9.73 ^a^
KL2	28.63 ^a^	2.90 ^a^	0.67 ^a^	2.98 ^a^	0.7	89.80 ^a^	0.122 ^a^	1.518 ^a^	0.146 ^a^	9.71 ^a^
KL3	26.93 ^b^	2.96 ^a^	0.73 ^a^	3.05 ^a^	2.39	90.96 ^a^	0.126 ^a^	1.564 ^a^	0.126 ^a^	9.75 ^a^
KL4	26.89 ^b^	2.98 ^a^	0.79 ^a^	3.08 ^a^	2.43	90.56 ^a^	0.133 ^a^	1.540 ^a^	0.150 ^a^	9.67 ^a^
*p* Values	<0.001	0.359	0.076	0.269	-	0.157	0.581	0.088	0.262	0.551

Note: Mean values of Chinese lacquer films’ properties followed by the same small superscript letters (^a,b^) within a group are not significantly different based on Fisher’s Protected LSD test at the 0.05 significance level.

## Data Availability

Data are contained within the article.

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
