# Peer review of "The Effects of Repeated Kurome Treatment on Chinese Lacquer and Its Film Properties"

_polymers, 2025, doi:10.3390/polym17111481_

Round 1

Reviewer 1 Report

Comments and Suggestions for Authors

The authors discussed the Kurome treatment of Chinese Lacquer and its film properties systematically. The design of the experiments and the results are interesting. Before the acceptance of the manuscript for publication, the authors need to revise the manuscript with clarifications for the following comments.

(1) The authors considered FTIR as one of the confirmation tool to discuss the results obtained in the present studies. Some of the major peaks obtained in FTIR spectra are not attributed. E.g.,the peak at around 3360 cm−1, which mainly due to the moisture present in the sample. As urushiol is a natural product and the subsequent treatment using water is still present in the sample, which is witnessed with peak at 3360 cm−1. In untreated sample, the water content is more as it appeared with a high intensity absorption peak. During the treatment, the sample was dried some extent and hence its absorption decreased. It is suggested to dry the sample (treated or untreated) at 100°C for 2 h followed by cooling to room temperature in a desiccator and then record the FTIR, surely the peak at 3360 cm−1 disappears.

(2) As the sample contains Urishiol, which is a 3-n-alk-(en)-yl-catechols, (alkane may contain 15 or 17 carbons with unsaturated bonds (i.e., C=C bonds), depending on the source of sample i.e., from which tree it has been taken), the -OH functional groups of the catechol also give an absorption at around 3350 cm−1, once it is ruled out as peak is due to water present in the bulk of the sample, the peak at around 3010 cm−1 can be attributed to the -OH of catechol.

(3) The peaks at 2925, 2855 cm−1 are not attributed, which may be due to aliphatic C-H and aromatic C-H respectively.

(4) As the authors rightly mentioned, the sample is undergoing oxidation on Kurome treatment which is also reflected in its IR spectra. As authors mentioned (line 199-200), the skeletal vibrations of C=C are weakening, maybe due to its involvement in polymerization process.

(5) Also, the catechol ring undergoes oxidation, forming an electrophilic o- quinone, which contains C=O on adjacent carbons on an aromatic ring, the peak at 1730 cm−1 is obtained for all the treated samples infers that the sample is undergoing oxidation and forming quinone during the treatment.

(6) The authors studied the light transmittance of the samples and are given in fig. 2. It is observed that the transmittance was not much appreciable change, it may be ±4% observed. The authors have not mentioned as if all the samples have of same thickness or not. The transmittance may vary with the thickness of the sample too. If they have the same thickness, how have they prepared?

Author Response

Journal: Polymers (ISSN: 2073-4360)

Manuscript ID: polymers-3663509

Type: Article

Title: The Effects of Repeated Kurome Treatment on Chinese Lac-quer and Its Film Properties

Authors: Jiangyan Hou, Yao Wang, Tianyi Wang, Guanglin Xu, Xinhao Feng and Xinyou Liu

ANSWER TO REVIEWER 1

Dear Reviewer,

We are grateful to you for the thorough review of our above contribution and the valuable comments and suggestions for improvement. We did carefully consider all your comments and did our best to follow them in the revision process of our paper. When this was not entirely possible, arguments were given.

A revised manuscript has been now submitted in two forms: with track changes for all modifications and without track-changes but highlighted changes (to facilitate reading and evaluation).

All the reviewers comments were numbered (Rx.y- where Rx- is the code of reviewer and y the corresponding number of its comment), so that you will find in the revised manuscript justification comments for each change. 

Please find below a copy of your Review report with all your suggestions and comments highlighted in red and our answers in black.

We do hope that the revised manuscript amended according to the input of the 2 reviewers, as much as this was possible, will meet the necessary standards for acceptance and publication.

Thank you again to you and the other reviewer for your effort, comments, constructive criticism and valuable advice for improving not only our current contribution but also our future research.

Sincerely yours,

Xinyou Liu, Corresponding’s authors

Reviewer 2 Report

Comments and Suggestions for Authors

The authors presented an interesting paper on the effect of multiple Kurome treatments on Chinese lacquer. The paper used many methods to determine the various properties of the resulting films, and the authors carefully approached the promlem of sample preparation and the exclusion of external factors. However, overall, the paper seems weak and insufficiently substantiated:

  1. In the introduction, the authors indicated that similar studies have been conducted previously, but primarily using a single Kurome treatment cycle (lines 53-54… Nonetheless, existing studies have primarily focused on single-round Kurome treatments…). The word “primarily” does not indicate that previously studies have been conducted on only one treatment cycle. How relevant is the study presented in the article, if further in the section with Results, the authors indicate that 2 treatment cycles are the most optimal. The authors should place more emphasis on the novelty of the work in the introduction.
  2. The article is quite small (about 3400 words, 11 pages, and 3 figures). In lines 191-193, the authors write "Future research should incorporate enzyme inhibition assays and gel permeation chromatography (GPC) to provide a more comprehensive mechanistic understanding of Kurome-induced modification". Why not supplement this article with such data right away to make it more complete and informative? It would also be possible to supplement the work with mechanical tests.
  3. Table 1 presents data on Viscosity, Surface dry time and Full curing time. It is evident (and the authors describe this) that with an increase in Kurome treatmentcycles, the viscosity increases noticeably. At the same time, Surface dry time and Full curing time decrease regularly, but in the last cycle (transition from KL3 to KL4) there is an increase in these indicators. Why is this happening? The authors' explanation of internal stresses is not very clear. If this is an exit to a plateau, then why does this not happen with viscosity?
  4. How do the high viscosity values for several Kurome treatment cycles correlate with the possibility of using such lacquer for coating? Will this contribute to an uneven (with lumps) varnish layer and, accordingly, lead to a deterioration in the appearance of the coating? Further in the article, this is indicated by the data in Table 2 and Fig. 2.
  5. The FTIR data look interesting, but it would be useful to supplement them, firstly, with the dependence of the carbonyl group band intensity on the treatment cycles (after normalizing the spectrum) to prove what is described in the text, and secondly, with a conditional reaction scheme with the destruction of some groups and the formation of others according to what the authors wrote.
  6. The presented SEM micrographs are too small and not of high enough quality. It is difficult to even estimate the scale markers from them, and the film surfaces do not look clear enough for analysis.

Author Response

Journal: Polymers (ISSN: 2073-4360)

Manuscript ID: polymers-3663509

Type: Article

Title: The Effects of Repeated Kurome Treatment on Chinese Lac-quer and Its Film Properties

Authors: Jiangyan Hou, Yao Wang, Tianyi Wang, Guanglin Xu, Xinhao Feng and Xinyou Liu

ANSWER TO REVIEWER 2

Dear Reviewer,

We are grateful to you for the thorough review of our above contribution and the valuable comments and suggestions for improvement. We did carefully consider all your comments and did our best to follow them in the revision process of our paper. When this was not entirely possible, arguments were given.

A revised manuscript has been now submitted in two forms: with track changes for all modifications and without track-changes but highlighted changes (to facilitate reading and evaluation).

All the reviewers comments were numbered (Rx.y- where Rx- is the code of reviewer and y the corresponding number of its comment), so that you will find in the revised manuscript justification comments for each change. 

Please find below a copy of your Review report with all your suggestions and comments highlighted in red and our answers in black.

We do hope that the revised manuscript amended according to the input of the 2 reviewers, as much as this was possible, will meet the necessary standards for acceptance and publication.

Thank you again to you and the other reviewer for your effort, comments, constructive criticism and valuable advice for improving not only our current contribution but also our future research.

Sincerely yours,

Xinyou Liu, Corresponding’s authors

Round 2

Reviewer 2 Report

Comments and Suggestions for Authors

The authors have made a number of changes to the manuscript and it has become better. It is a pity that the material was not revised more substantially in accordance with the recommendations (for example, the authors never provided a reaction scheme to illustrate the ongoing cross-linking processes that they describe in the FTIR data). Repairs in the laboratory are great, but they should not potentially be a reason for the impossibility of finalizing the article. I wish the authors to conduct the promised series of additional studies in the future. In general, the presented version of the manuscript can be published if the editor has no questions about the volume of the text.